# A Robust and Efficient Loop Closure Detection Approach for Hybrid Ground/Aerial Vehicles

**Yutong Wang** [1,2], **Bin Xu** [1,2,*], **Wei Fan** [1,2] and **Changle Xiang** [1,2]

1   School of Mechanical Engineering, Beijing Institute of Technology, Beijing 100081, China
2   China and Beijing Institute of Technology Chongqing Innovation Center, Chongqing 401147, China
*   Correspondence: bitxubin@bit.edu.cn

**Abstract:** Frequent and dramatic viewpoint changes make loop closure detection of hybrid ground/aerial vehicles extremely challenging. To address this issue, we present a robust and efficient loop closure detection approach based on the state-of-the-art simultaneous localization and mapping (SLAM) framework and pre-trained deep learning models. First, the outputs of the SuperPoint network are processed to extract both tracking features and additional features used in loop closure. Next, binary-encoded SuperPoint descriptors are applied with a method based on Bag of Visual Words (BoVW) to detect loop candidates efficiently. Finally, the combination of SuperGlue and SuperPoint descriptors provides correspondences of keypoints to verify loop candidates and calculate relative poses. The system is evaluated on the public datasets and a real-world hybrid ground/aerial vehicles dataset. The proposed approach enables reliable loop detection, even when the relative translation between two viewpoints exceeds 7 m or one of the Euler angles is above 50°.

**Keywords:** hybrid ground/aerial vehicles; visual SLAM; loop closure detection

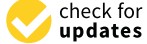



## 1. Introduction

Recently, there has been some considerable interest in hybrid ground/aerial vehicles [1–3]. These unique vehicles are generally designed to combine the advantages of both terrestrial and aerial modes by moving on the ground to save energy and flying when terrain constraints do not allow. Hybrid ground/aerial vehicles are excellent alternatives in applications such as search and rescue or military surveillance due to their flexibility. To achieve autonomous navigation in unknown, cluttered, and GPS-denied environments, hybrid ground/aerial vehicles are often equipped with visual sensors, which are compact and lightweight [4]. Based on visual information, the visual simultaneous localization and mapping (SLAM) technique can estimate ego-motion and reconstruct the structure of the environment. In the past decades, SLAM has been well-developed for unmanned aerial vehicles (UAVs) or unmanned ground vehicles (UGVs) [5–10], while specific challenges brought by the characters of hybrid ground/aerial vehicles are rarely studied. Among the challenges, loop closure detection failures caused by frequent and dramatic viewpoint changes are crucial, as is shown in Figure 1.

In this paper, the loop closure detection problem in visual SLAM for hybrid ground/aerial vehicles is investigated. Loop closure detection is a vital component of state-of-the-art visual SLAM systems. By recognizing the revisiting places correctly, loop closure is not only beneficial to reduce the accumulated position errors but also useful for relocalization in a previous map. The Bag of Visual Words (BoVW)-based methods, such as FAB-MAP [11,12] and DBoW [13], are by far the most used techniques for loop closure in visual SLAM methods, such as LSD-SLAM [14], ORB-SLAM3 [15] and VINS-Mono [16]. Most BoVW-based methods use handcrafted keypoint features, such as SIFT [17], SURF [18], ORB [19] and BRIEF [20]. However, handcrafted feature descriptors extracted from images of the same place, captured at ground and air view, can vary drastically. It can cause errors in both image retrieval and

image matching, which are critical in loop closure detection. Thus, the traditional approaches are inadequate for hybrid ground/aerial vehicles with a wide range of viewpoints.

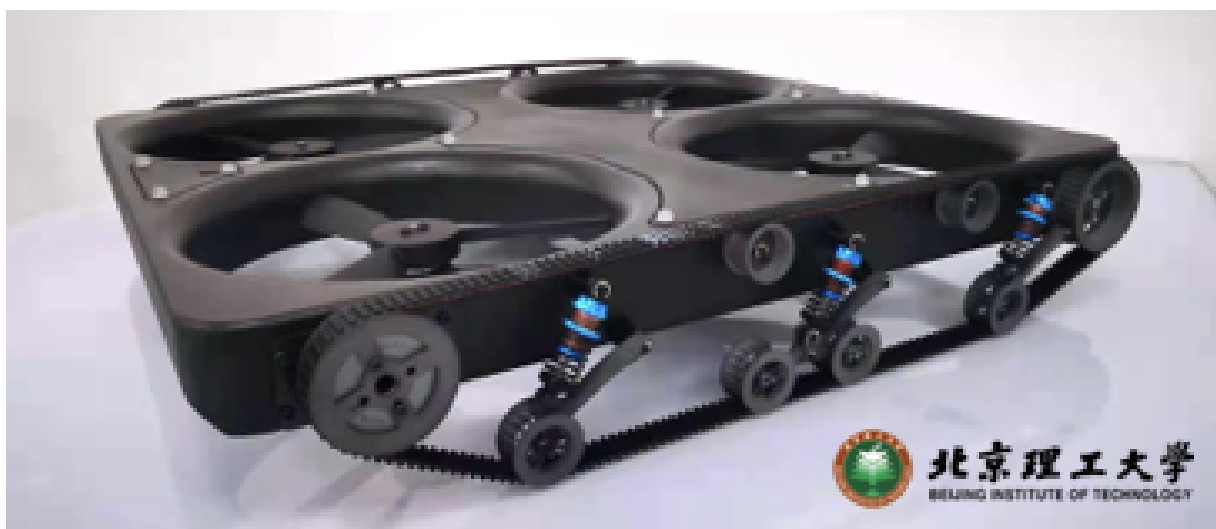

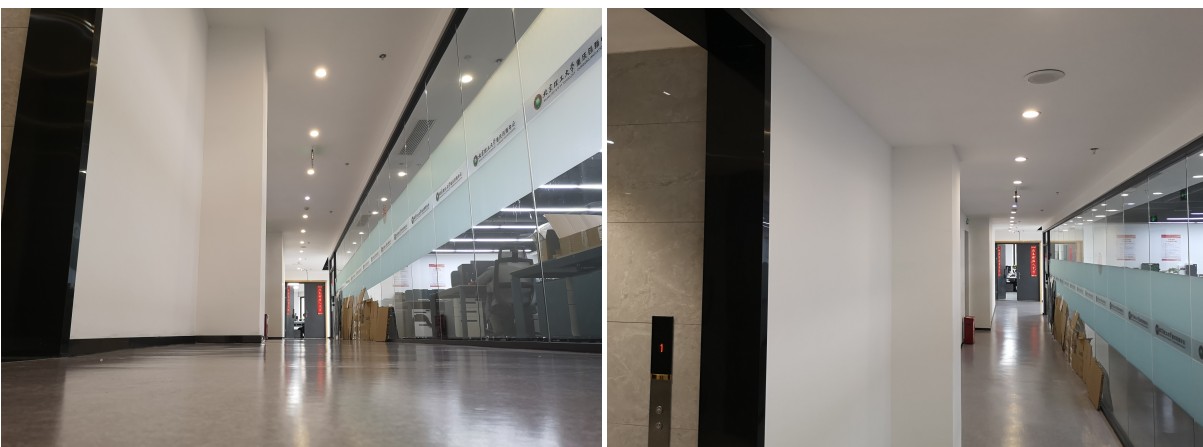

**Figure 1.** It is challenging for hybrid ground/aerial vehicles to close loops, as they can revisit a place with different perspectives. The **top picture** shows a self-developed hybrid ground/aerial vehicle with four ducted fans for aerial mode and two continuous tracks for terrestrial mode (The non-English characters on the picture represents for the Chinese name of Beijing Institute of Technology). The **bottom row** illustrates the remarkably distinct appearances of an image pair captured at ground-level and aerial views.

With the development of deep convolutional neural networks (DCNNs), using learning-based local features to improve the loop closure pipeline is a new trend [21,22]. Though having shown the potential for robustness against view changes, most learning-based methods suffer from slow feature extraction and slow querying. Recently, experimental results of SuperGlue [23] have illustrated that combining SuperGlue with SuperPoint [24] enables matching extreme wide-baseline image pairs accurately and efficiently. However, integrating the deep learning models into the appearance-based loop closure detection approaches is also non-trivial. At first, strategies to extract features in practical SLAM frameworks are different from offline image processing. Secondly, the SuperPoint descriptors are difficult to be clustered in BoVW-based methods, as the descriptors are high-dimensional, float formatted and hardly explainable.

To address the problems mentioned above, we present a robust and efficient loop closure detection approach for hybrid ground/aerial vehicles. To combine pre-trained SuperPoint models with state-of-the-art SLAM frameworks, the outputs of SuperPoint are processed to

extract both tracking and additional features. To improve the efficiency of keyframe retrieval, the SuperPoint descriptors are encoded and utilized in a BoVW model. Moreover, SuperGlue is applied to find precise feature correspondences with SuperPoint descriptors.

The main contributions of this paper are the following:

- We utilize an encoding method for SuperPoint descriptors in a BoVW model to train a compact vocabulary for speeding up keyframe retrieval.
- We propose a novel loop closure approach by embedding learning-based models into state-of-the-art SLAM frameworks for hybrid ground/aerial vehicles.
- We evaluate the performance on EuRoC [25] datasets and the real-world data collected by a hybrid ground/aerial vehicle.
- We provide a direct and high-performance replacement for the loop closure algorithm in the popular VINS-Fusion pipeline to handle cases with large viewpoint changes. The code is released at https://github.com/yutongwangBIT/SP-Loop (accessed on 10 February 2023) to further work in the field by the broader research community.

The remainder of this paper is structured as follows. In Section 2, we review published work related to viewpoint change issues and efficient loop closure detection approaches. The process of offline vocabulary training with the proposed descriptor encoding method and the proposed online loop closure detection algorithm is presented in Section 3. Section 4 introduces the implementation details. In Section 5, experimental results are presented, compared with the state-of-the-art and discussed. Finally, Section 6 is left to the conclusions and to describe the future work.

## 2. Related Work

### 2.1. Viewpoint-Invariant Image Matching

Changes between ground and aerial modes of the hybrid ground/aerial vehicles lead to the requirement of robustness against perspective changes. Although loop closure detection for hybrid ground/aerial vehicles is, so far, unexplored, the viewpoint issues have been investigated in some other applications, such as geo-localization of the UGVs in GPS-denied environments and collaborative localization of UAVs/UGVs [26]. The methods can be divided into three main categories: structure-based, semantic and learning-based methods. Structure-based methods are usually applied to deal with the geo-localizing problem in urban scenes where the building facades have self-similar structures [27,28]. Some researchers have used object semantics from images to localize [29–31], but some objects do not have a precise bounding box to model semantic entities.

Recent studies on deep learning for image matching have illustrated the advanced abilities of deep local features. UCN [32] and DeepDesc [33] have achieved fair results for image matching with the extracted descriptors, but they do not perform any interest point detection. On the contrary, despite the fact that LIFT [34] can detect keypoints, estimate orientations and compute descriptors, it has been proven to not be competitive with hand-engineering features in a real-life application [35]. The proposal of SuperPoint, an interest point detector and description extractor trained by the unsupervised fully-convolutional neural network architecture, has attracted researchers' attention [24]. Further, the Graph Neural Networks (GNNs)-based SuperGlue has been introduced to improve the ability to find feature correspondences [23]. It has been shown that SuperGlue achieves advanced results in challenging environments, especially when combined with SuperPoint. Moreover, both SuperPoint and SuperGlue can perform on a modern GPU (Graphics Processing Unit) in real-time.

### 2.2. Efficient Loop Closure Detection Approaches

To be used in real-time visual SLAM systems, loop closure detection should be efficient. There are plenty of investigations aiming at improving its efficiency based on handcrafted local features. Since the 21st century, BoVW-based approaches have been widely used in practical SLAM systems. FAB-MAP and FAB-MAP 2.0 [11,12] represent images with a bag of words based on SURF, and a Chow–Liu tree has been applied to learn the words'

covisibility probability offline. Moreover, DBoW [13] has been boosted with binary BRIEF descriptors and a k-dimensional (k-d) tree. Most handcrafted feature-based methods require a stage of geometrical check due to the low recall rate and lack of matching correctness.

In contrast to handcrafted local features, efficiently using learning-based features in loop closure detection is an open question. Yue et al. [36] have used SuperPoints for place recognition based on an incremental BoVW approach with a graph verification process. GCN [37] has evaluated a learned keypoint and descriptor detector using a naive SLAM system, including loop closure with a BoVW library. Although the studies mentioned above have integrated learned features into the fast BoVW models, none have tested the algorithms in real-time datasets. Based on GCN, GCNv2 has been designed to suit the real-time ORB-SLAM2 by training networks for binarized feature descriptors and keypoint detectors [38]. GCNv2 has demonstrated the potential of integrating pre-trained deep learning models into real-time systems. Inspired by the above analysis, our research focuses on providing a more robust and efficient loop closure detection approach with the combination of simplified SuperPoint features and BoVW models.

## 3. Methods

The whole process is divided into two parts, namely offline vocabulary trainingand online loop closure detection. The former describes the approach of combing the deep-learned descriptors with the traditional BoW methods, while the latter introduces the proposed loop closure algorithm and its application in a state-of-the-art SLAM framework.

### 3.1. Offline Vocabulary Training

In this section, we first present the overview of the offline process based on a BoVW-based approach. The method of encoding the original deep-learned descriptors to speed up the offline training is then introduced.

#### 3.1.1. Offline Process Overview

To detect revisited places for loop closure efficiently, we use the DBoW [13] approach with binary-encoded SuperPoint features. DBoW constructs a hierarchical visual dictionary containing lots of visual words through clustering local feature descriptors extracted from a large-scale image training dataset. Thus, before its application in the online loop closure system, we create the visual vocabulary offline by discretizing the descriptor space into visual words.

The pipeline of offline training is shown in Figure 2. To obtain a rich set of features, we prepare sufficient training images from different datasets, such as ICL-NIUM [39] and TartanAir [40]. It is worth noting that the training images are independent of those processed online later. With the open-released pre-trained weights of SuperPoint net, the input tensors that are converted from the original images are first encoded. Then the encoded tensors are decoded by the interest point decoder and descriptor decoder separately. The descriptors used for training are extracted according to the keypoints from the interest point decoder. Instead of using the original descriptors with float values, we discretize a binary descriptor space to create a more compact vocabulary. The details of binary encoding are described later. The binary SuperPoint descriptors are utilized in the DBoW process to build the tree-structured vocabulary. In the DBoW process, the descriptors are first clustered into $k$ groups by executing k-medians clustering with k-means++ [41]. The $k$ clusters constitute the first level of the vocabulary tree nodes, while the subsequent levels are built by repeating the clustering operation on each node. Finally, a tree with nodes and leaves (visual words) is constructed. Moreover, a direct index storing the features of each image and an inverse index storing a list of images for each word are also generated to speed up querying and verifying loop candidates later.

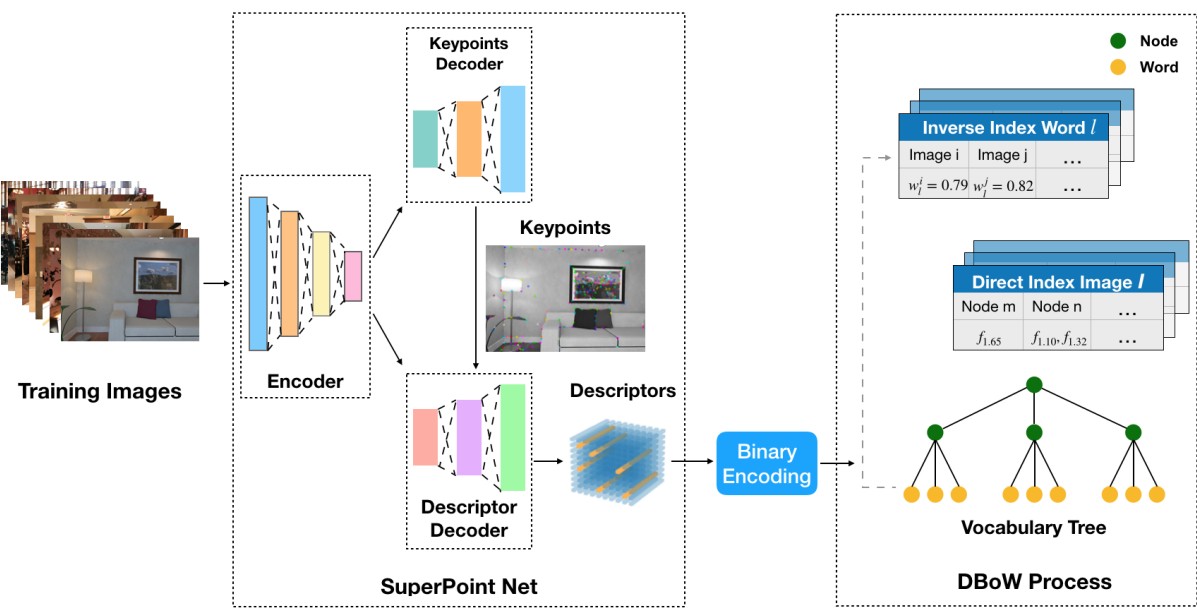

**Figure 2.** A diagram illustrating the process of offline vocabulary training.

### 3.1.2. Binary Encoding

The DBoW approaches in the state-of-the-art visual SLAM frameworks prefer utilizing handcrafted binary features. In the training process, they can construct compact vocabularies. In the process of online loop closure, the keyframe query can also be extremely efficient with binary descriptors, as they can be processed and clustered fast. By contrast, the deep-learned features, such as SuperPoint descriptors, are high-dimensional, concentrated and in float format. We analyzed the value of SuperPoint descriptors extracted from several independent images statistically. A histogram of the collected descriptor values is shown in Figure 3. There is a total of 1675 descriptors with 256 dimensions involved. The colors on the histogram are randomly selected and used to distinguish the dimensions. As can be observed in Figure 3, the values of the descriptors are normally distributed with an expected value of 0 and a standard deviation of 0.07. Moreover, this distribution does not change with dimensions, as the colors in the figure are evenly distributed. It is clear that values like this are hard to be discretized into clusters of the tree-structured vocabulary.

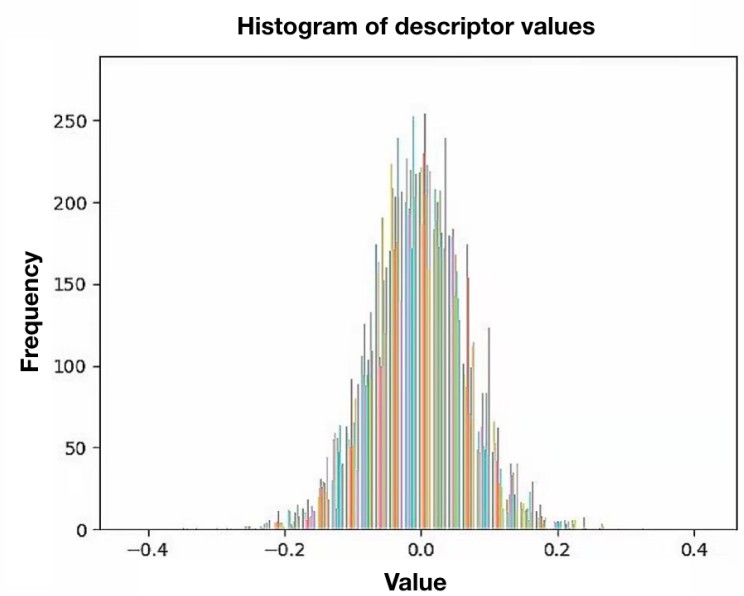

**Figure 3.** The histogram of extracted SuperPoint descriptor values.

Arroyo et al. [42] binarized CNN features to make topological localization more robust, while Hafez et al. [43] encoded the deep-learned features to speed up querying for visual place recognition. Inspired by them, we employ encoded descriptors to train the vocabulary and later the online keyframe querying. According to the above data analysis of SuperPoint descriptor values, we propose a practical formula to encode the values in a descriptor vector $\mathbf{v}$:

$$\mathbf{v}_i = \begin{cases} 1, & \text{if } \mathbf{v}_i \geq 0 \\ 0, & \text{otherwise} \end{cases} \tag{1}$$

where $\mathbf{v}_i$ denotes the $i$th value of the vector.

After being binarized, each descriptor becomes an appreciably more compact vector consisting of bits. To illustrate that the binary SuperPoint descriptors have similar performance as the original ones, we display the comparison of ground/air image matching results in Figure 4. As can be seen in Figure 4, there are several apparent incorrect matches in the pictures with BRIEF and ORB descriptors, while all of the matches are true in the picture with the original SuperPoint descriptors. As for the result of our binary SuperPoint descriptors, only one false match can be observed. The accuracy of feature matching can indicate the distinctiveness of the descriptors. Thus, the results of matching demonstrate that the reduction in the distinctiveness caused by our process of encoding is negligible.

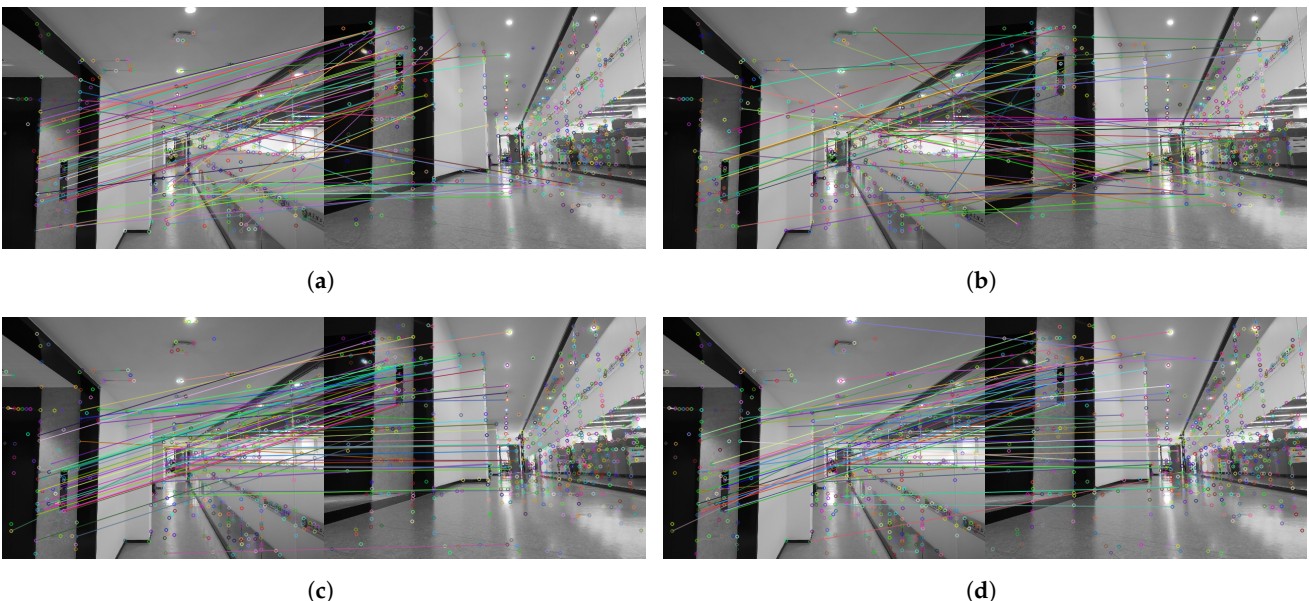

**Figure 4.** Comparison of image matching results. (**a**) With BRIEF descriptors. (**b**) With ORB descriptors. (**c**) With the original SuperPoint descriptors. (**d**) With the binary SuperPoint descriptors.

In total, the proposed encoding technique has a good compromise between effectiveness and distinctiveness. On the one hand, not only the training process but also the querying in the online process is sped up. On the other hand, the performance of binarized SuperPoint descriptors and the original ones are almost identical, which is noticeably superior to handcrafted descriptors.

### 3.2. Online Loop Closure Detection

In this section, we first present the overview of the developed online process based on a state-of-the-art SLAM framework. Next, the technical details of feature extraction, keyframe retrieval, feature matching and pose computing are introduced sequentially.

### 3.2.1. Online System Overview

Our online system is developed on the basis of the state-of-the-art VINS-Fusion [16], which contains visual-inertial odometry (VINS-VIO), pose graph optimization (VINS-PGO)

and loop closure detection (VINS-Loop). In this work, VINS-VIO and VINS-PGO are adopted, while our proposed loop closure detection approach is used to replace VINS-Loop to improve robustness against viewpoint changes. The flow diagram illustrating the pipeline of online loop closure is shown in Figure 5.

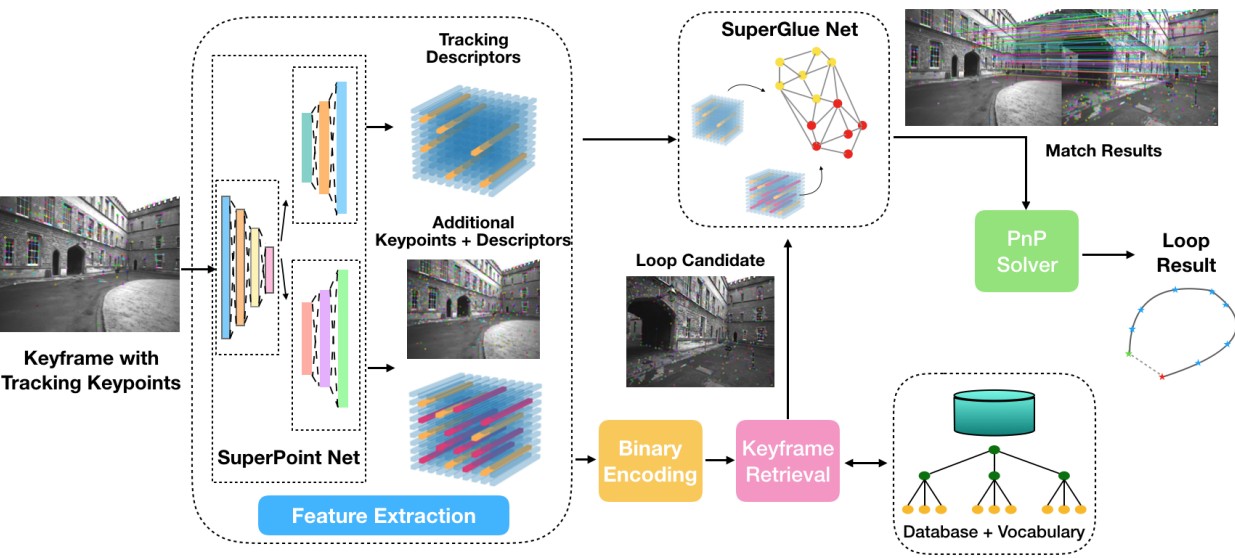

**Figure 5.** A diagram illustrating the pipeline of online loop closure.

Every new coming keyframe with tracking features from VINS-VIO is processed. The same as in the offline training process, we use the open-released SuperPoint weights in our real-time online system since they are lightweight and efficient [24]. There are two parts of features after the feature extraction process: tracking features that are already selected in VINS-VIO and additional features extracted in the loop closure process to improve the performance of place recognition. The tracking features are used directly in the feature-matching process. The additional descriptors are binarized as in the offline training to query a loop candidate from the keyframe database based on the offline-trained visual vocabulary. The SuperGlue net is applied to find feature correspondences between the tracking features of the current keyframe and the additional features of the retrieved keyframe candidate. While tracking features of the current keyframe can provide 3D position estimates for the pose computation, the additional features can increase the number of matches. The output of a SuperGlue net contains matches and confidence. Only the matches with confidence above a set confidence threshold are selected. If the number of matching pairs is sufficient, the loop candidate is accepted. Finally, the relative pose between the current keyframe and the loop candidate is calculated by the Perspective-n-Point (PnP) method [44]. The loop information, including the index of the candidate and the relative pose, is stored. During a pose graph optimization, the loop information is added to the pose graph as additional constraints. The more reliable constraints are, the more accurate estimation results can be obtained.

### 3.2.2. Feature Extraction

Each new coming keyframe contains a corresponding image and tracking features. The original image **I** (width $W$ and height $H$) is converted to grayscale and resized to an input tensor of shape $W' \times H'$, which is defined by the pre-trained SuperPoint nets. For example, if the SuperPoint weights are pre-trained with setting the resolution of $640 \times 480$, then $W' = 640, H' = 480$ here. Higher resolution contributes to the increasing distinctive of the features, but it also leads to higher computational demand and larger memory space. In practice, $640 \times 480$ is sufficient to extract features. Then the input tensor of shape $W' \times H'$ is forwarded by the pre-trained SuperPoint nets. The output tensors are the score tensor

$\mathbf{S} \in \mathbb{R}^{H' \times W'}$ and the descriptor tensor $\mathbf{D} \in \mathbb{R}^{1 \times L \times \frac{W'}{8} \times \frac{H'}{8}}$, where $L$ is the length of a single descriptor (e.g., $L = 256$). Tracking descriptors are extracted from $\mathbf{D}$ according to the image coordinates of tracking features. The number of tracking features is bounded in VINS-VIO with a maximum value (100–300), which ensures the real-time performance of VINS-VIO but is not sufficient for image retrieval in loop closure. To increase the recall rate, 500 to 1000 additional features are detected and described.

Once the image is processed and forwarded by the pre-trained SuperPoint nets, tracking descriptors and additional features are extracted from the SuperPoint output tensors $\mathbf{S}$ and $\mathbf{D}$. The original feature extraction methods in SuperPoint cannot be directly used in the proposed system. The tensors are processed to extract the two kinds of descriptors, as illustrated in Figure 6.

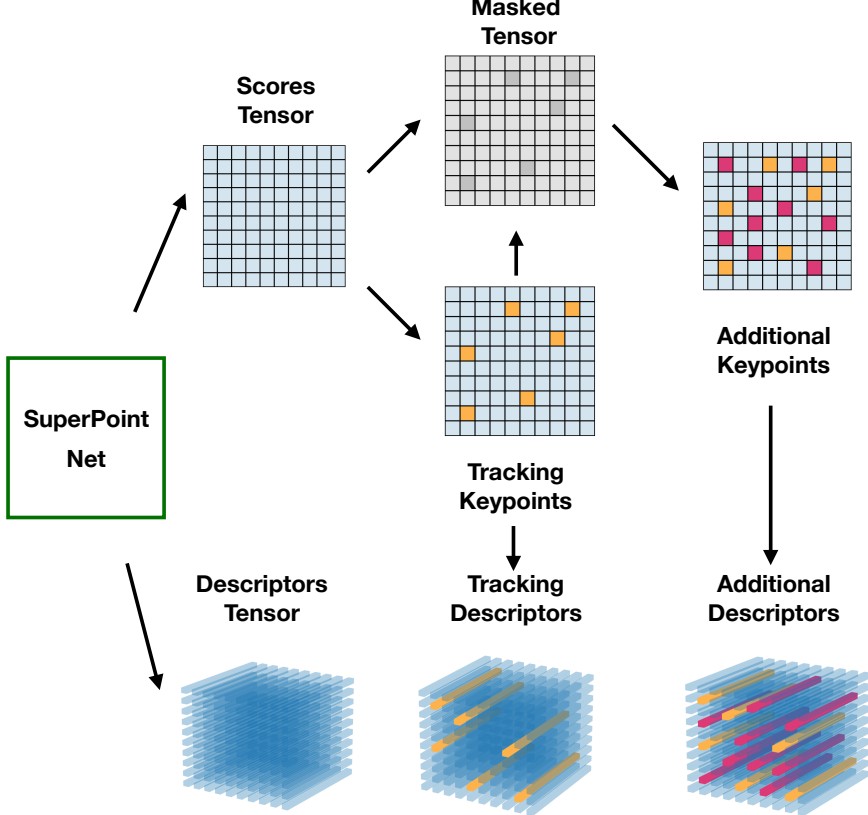

**Figure 6.** The data flow diagram of feature extraction. The yellow blocks and bars represent the tracking keypoints and descriptors, while the red blocks and bars are the addtional keypoints and descriptors.

Tracking features from VINS-VIO include the image coordinates of keypoints during tracking. Let $\mathcal{A} := \left\{ \mathbf{f}_k^{\mathrm{w}} \right\}_{k=1}^{n}$ be a set of $n$ known tracking keypoints, where $\mathbf{f}_k^{\mathrm{w}} := (u_k \cdot H'/H, v_k \cdot W'/W)$ is the resized pixel location of a keypoint with 2D coordinates $(u_k, v_k)$. Then, a set of corresponding tracking descriptors $\left\{ \mathbf{d}_k^{\mathrm{w}} \right\}_{k=1}^{n}$ is computed using the known tracking keypoints $\mathcal{A}$ and descriptors tensor $\mathbf{D}$.

After that, additional features are extracted from the masked tensors. As the tracking features cannot be selected twice; the score tensor is masked with known pixels. The masked tensor $\mathbf{S}' \in \mathbb{R}^{H' \times W'}$ is constructed as:

$$\mathbf{S}' = \left\{ S'_{i,j} \mid S'_{i,j} = \left\{ \begin{array}{ll} \mathbf{S}_{i,j}, & \text{if } (i,j) \notin \mathcal{A} \\ 0, & \text{else} \end{array} \right. \right\}, \tag{2}$$

where $i \in \{1, \ldots, H'\}$, $j \in \{1, \ldots, W'\}$. To decrease feature redundancy, local maximums, whose score values are greater than all neighbors, are searched by Non-Maximum Suppression (NMS). The process is:

$$\mathbf{S}'' = g(\mathbf{S}', d), \tag{3}$$

where $g(.)$ represents the NMS function, and $d$ denotes a minimum NMS distance. The processed score tensor $\mathbf{S}''$ is then constructed to $\mathbf{S}'''$ with a threshold of scores $t$:

$$\mathbf{S}''' = \left\{ S'''_{i,j} \mid S'''_{i,j} = \left\{ \begin{array}{ll} \mathbf{S}''_{i,j}, & \text{if } \mathbf{S}''_{i,j} > t \\ 0, & \text{else} \end{array} \right. \right\}, \tag{4}$$

where $i \in \{1, \ldots, H'\}$, $j \in \{1, \ldots, W'\}$.

Then, $m$ largest nonzero elements $\{(u'_k, v'_k)\}_{k=1}^m$ in $\mathbf{S}'''$ are selected. According to the elements and $\mathbf{D}$, additional descriptors $\{\mathbf{d}_k^a\}_{k=1}^m$ are computed. The set of additional keypoints is $\{\mathbf{f}_k^a\}_{k=1}^m$, where $\mathbf{f}_k^a := (u'_k \cdot H/H', v'_k \cdot W/W')$ is the back resized pixel location.

### 3.2.3. Keyframe Retrieval

A loop candidate is recalled based on the offline-trained vocabulary and a keyframe database. The database stores processed vectors of all passing keyframes. When a keyframe $\mathrm{K}_t$ is processed, its binary descriptors are converted into a vector $\mathbf{v}_t$ of weighted visual words according to the vocabulary. The vector $\mathbf{v}_t$ searches vectors $\{\mathbf{v}_{t_i}\}_{i=1}^{t-T}$ in the database, i.e., $T$ latest keyframes are ignored for loop candidates. It results in a list of matching candidates $\langle v_t, v_{t1} \rangle, \langle v_t, v_{t2} \rangle, \ldots$, associated with the matching scores $s(v_t, v_{tj})$. Candidates that do not reach a minimum score threshold $s_c$ are rejected. If no candidate meets the requirements, $\mathrm{K}_t$ is skipped after being added to the database. Otherwise, the oldest candidate $v_c$ among the accepted candidates is chosen, as it is assumed that estimates of earlier appeared keyframes are more reliable. With the obtained index, $tc$, the candidate keyframe $\mathrm{K}_c$ is recalled. $\langle \mathrm{K}_t, \mathrm{K}_c \rangle$ is thus a candidate pair for feature matching.

### 3.2.4. Feature Matching and Pose Computing

SuperGlue is utilized to find feature correspondences correctly, even if the image baseline of candidate pairs is extremely wide. The SuperGlue net learns priors over geometric transformation and regularities through training from image pairs. It leverages both spatial relationships of the keypoints and their descriptors. Here, correspondences are found between tracking features of the current keyframe $\mathrm{K}_t$ and additional features of the loop candidate $\mathrm{K}_c$. While the former can provide 3D position estimates in the world frame for the pose computation, the latter can increase the number of matches. Next, an input tensor $\mathbf{M}$ is constructed with keypoints, response values and descriptors of the features. $\mathbf{M}$ is then forwarded by an attentional graph neural network and an optimal matching layer. The output tensors include matches and confidences. Only the matches with confidences above a set confidence threshold $c_t$ are selected. After that, $L$ matching pairs $\{\langle \mathbf{f}_{t_l}, \mathbf{f}_{c_l} \rangle\}_{l=1}^L$ of features and their confidences $\mathbf{c} \in (c_t, 1]^L$ are calculated.

A candidate pair with an insufficient number of correspondences is rejected. Otherwise, the loop is detected. Then, a relative transformation $\mathbf{T}_{tc}$ between $\mathrm{K}_t$ and $\mathrm{K}_c$ is computed. A direct Perspective-n-Point (PnP) method [44] is performed to estimate the pose using 3D points and 2D projections. Unlike VINS-Loop, where RANdom SAmple Consensus (RANSAC) [45] iterations are demanded to reject outliers, a simple and efficient PnP process can provide accurate estimates in our work. Three-dimensional position estimates of matching features of $\mathrm{K}_t$ and the corresponding 2D keypoints locations of $\mathrm{K}_c$ are the inputs of PnP. Finally, loop information, including transformation $\mathbf{T}_{tc}$, is stored in $\mathrm{K}_t$. During a pose graph optimization operation, $\mathbf{T}_{tc}$ is added to the pose graph as additional constraints. The more reliable constraints are, the more accurate results are obtained.

## 4. Implementation Details

We implement the proposed algorithm with C++ and ROS (Robot Operating System) as VINS-Fusion. Normally, the deep learning models are trained and utilized in Python, the open-sourced pre-trained SuperPoint and SuperGlue weights are as well. To embed our loop closure algorithm into VINS-Fusion, we first deployed the Pytorch models using the Torch [46] C++ interface. In detail, the weights were converted to TorchScript, which can be loaded by C++. For the DBoW model, we use the open-source improved version DBoW3 library. The extracted and processed SuperPoint descriptors replaced the handcrafted BRIEF or ORB descriptors in the library. To achieve better real-time performance, we deployed the processes using SuperPoint and SuperGlue weights on the GPU while other computations are run on the CPU (Central Processing Unit).

We tested the algorithm on a laptop with a NVIDIA RTX 3080 and an Intel i9-10885H CPU. On the GPU, it takes, on average, around 20 ms for SuperPoint extraction and about 60 ms for SuperGlue, respectively. On the CPU, keyframe query from the database costs about only 3 ms. In total, the proposed loop closure detection can process keyframes at over 10 FPS on our device. It indicates that our implementation can run in real time with deep-learned models on a modern computer.

## 5. Experimental Results

We evaluate the different aspects of our proposal in the following. In Section 5.1, the ability of the proposed binary SuperPoint descriptors in image retrieval is analyzed. Next, the proposed loop closure detection algorithm is compared with VINS-Loop on public datasets in Section 5.2. We then present the results of a real-world experiment conducted with our quad ducted-fan hybrid ground/aerial vehicle in Section 5.3.

### 5.1. Descriptor Effectiveness

To investigate the effectiveness of the proposed binary SuperPoint descriptors, they are compared with the original SuperPoint, ORB and BRIEF descriptors using the DBoW3 library. Experiments were conducted on a public dataset and a self-collected ground/air dataset. Visual vocabularies of all descriptors were trained using 13,975 images of TartanAir [40] with a branching factor of 10 and depth levels of 6, resulting in $10^6$ = 100,000 visual words. Term frequency-inverse document frequency (tf-idf) [13] was the weighting type, and L1-Norm was the scoring type.

#### 5.1.1. Performance on Dataset

For the test on the public dataset, 261 images from over 45,000 images of the New College dataset [47] have been selected manually. Aiming at increasing the difficulty of image retrieval, the overlapping area between each image pair does not exceed 80%.

The precision-recall curves of the proposed binary SuperPoint, the original SuperPoint, ORB and BRIEF descriptors are illustrated in Figure 7. Note that the precision axis begins at 0.6. The first remark is that the area under the curve (auc) of the original SuperPoint descriptors is a lot lower than the others. It indicates the necessity of encoding. The figure also reveals that binary SuperPoint outperforms both BRIEF and ORB descriptors. At 100% precision, the proposed descriptors achieve approximately 80% recall rate, while the other two handcrafted descriptors can barely reach around 40%.

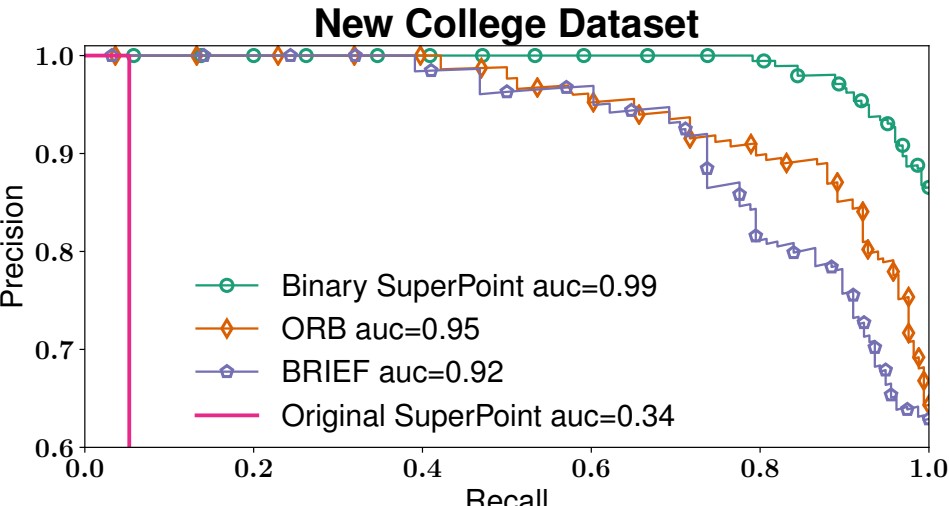

**Figure 7.** Precision-recall curves of binary SuperPoint, ORB, BRIEF and original SuperPoint descriptors on the New College dataset [47].

5.1.2. Influence of Viewpoint Difference

To better illustrate the reliability of the proposed binary descriptors under viewpoint change, 474 ground-level images and 467 air-level images have been collected. There are two kinds of experiments to be compared. The first one uses only ground-level images. The other experiment is called cross-verification. All air-level images are stored in a database, from which each ground-level image recalls an image.

The precision-recall matrices of binary SuperPoint, ORB and BRIEF descriptors are shown in Figure 8. Generally, curves with the term "(cross)" are lower than the curves of the first experiment. It demonstrates the difficulty of loop closure detection under perspective changes. It is also observed that the curve of binary SuperPoint in cross-verification has an area of 0.966 under the curve, higher than ORB (0.948) and BRIEF (0.896). The precision of binary SuperPoint in cross-verification at a recall rate of 100% is about 90%, the highest among all curves. It supports that using binary SuperPoint is more reasonable than handcrafted descriptors for image retrieval problems, especially on our hybrid ground/aerial vehicles.

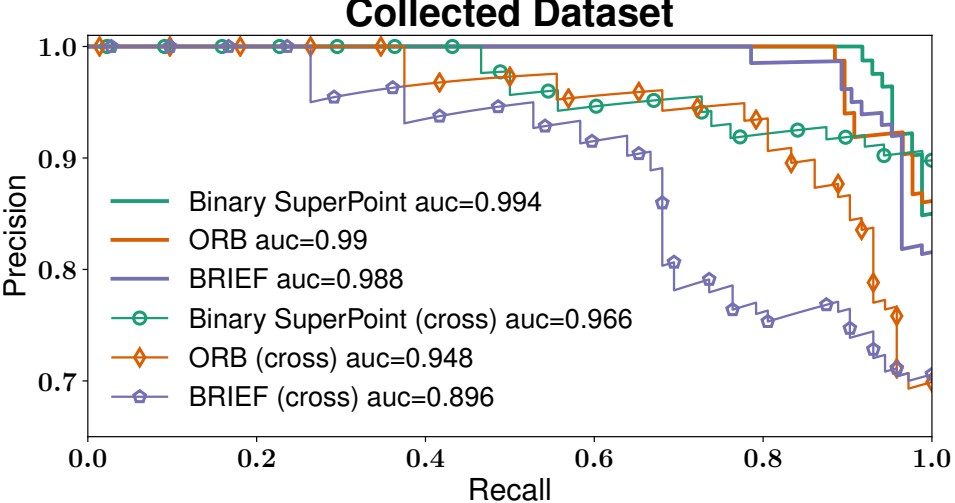

**Figure 8.** Precision-recall curves of binary SuperPoint, ORB, BRIEF and original SuperPoint descriptors on the collected dataset with ground/air images.

*5.2. Online Experiments on Datasets*

EuRoC presents visual-inertial datasets collected on-board a Micro Aerial Vehicle (MAV), providing 11 sequences recorded in two different rooms and a large industrial environment [25]. In our experiments, images from the left camera and IMU measurements were the inputs of VINS-VIO. The performance is compared with VINS-Loop, as both inputs are from the same VINS-VIO. First, we report the localization performance on every single EuRoC sequence. Secondly, we analyze the influence of viewpoint difference by setting the minimum relative pose to close the loop so that the extremely outstanding advantages of the proposed system on ground/aerial vehicles are proved. Moreover, to demonstrate the potential of our method for the collaborative SLAM with multi-robot system, we also show multi-session results in three rooms.

5.2.1. Single-Session Performance

We evaluate the localization accuracy by an absolute trajectory error (ATE), which is given by the difference between the estimated trajectory and ground truth with alignment. In detail, the estimated poses are associated with ground-truth poses using the timestamps as a pre-processing step. The maximally allowed time difference for matching entries is set as 0.02, which is an approximate value observed in practice. Based on this association, the true and the estimated trajectory are aligned using singular value decomposition. Finally, the difference between each pair of poses is computed. The output root-mean-square error (RMSE) [48] of all EuRoC sequences are listed in Table 1. Our report values are the median after five executions. It can be observed that the proposed algorithm outperforms the VINS-Loop in the majority of cases. For example, in sequence V2_02_medium, the error of the proposed system is 0.054, 0.2 and 0.04 m lower than VIO and VINS-Loop, respectively. In some cases, such as MH_02_easy and MH_04_difficult, there was an unapparent enhancement. The accuracy of the proposed approach is lower than VINS-Loop only in the V1_03_difficult sequence. In these cases, where our system has insignificant improvements, VINS-Loop also closes loops frequently, as the MAV revisited very similar places multiple times during the flights. Meanwhile, the camera view in the machine hall is normally broader than V1 and V2 rooms, so that we observe more obvious improvements in the latter environment.

**Table 1.** RMSE in EuRoC datasets in meters. The best result is highlighted in **bold**.

| Sequence | VINS-VIO | VINS-Loop | Proposed |
|:---:|:---:|:---:|:---:|
| V1_01_easy | 0.068 | 0.044 | **0.042** |
| V1_02_medium | 0.093 | 0.044 | **0.034** |
| V1_03_difficult | 0.133 | **0.080** | 0.082 |
| V2_01_easy | 0.065 | 0.049 | **0.038** |
| V2_02_medium | 0.246 | 0.093 | **0.054** |
| V2_03_difficult | 0.186 | 0.11 | **0.10** |
| MH_01_easy | 0.22 | 0.075 | **0.070** |
| MH_02_easy | 0.08 | 0.044 | 0.044 |
| MH_03_medium | 0.132 | 0.074 | **0.068** |
| MH_04_difficult | 0.292 | 0.10 | 0.10 |
| MH_05_difficult | 0.228 | 0.099 | **0.09** |

5.2.2. Influence of Viewpoint Difference

In order to demonstrate the outstanding advantages of our system under viewpoint variation, a minimum relative pose ($\min_t$ and $\min_\psi$) between a loop candidate pair is set. The relative translation $t_{ij}$ and yaw angle ($\psi_{ij}$) can be calculated from the estimated relative pose, which is obtained by the PnP solver. The two parameters can indicate the significance of viewpoint change. Thus, if the loop candidates with close viewpoints of the current keyframes are ignored, we can find the limit of difference in viewpoints that our system can withstand.

In the experiments, loop candidates are accepted as loops only if $|t_{ij}| > \min_t$ or $|\psi_{ij}| > \min_\psi$, and a list of thresholds pairs $\min_t \backslash \min_\psi$ is set. From all EuRoC sequences, we select sequence MH_04_difficult, on which our system has no significant improvement compared with VINS-Loop. The results are shown in Table 2, where the corresponding loop numbers and absolute translation RMSE of our system and VINS-Loop are compared. As expected, the more critical the constraints are, the more considerably the proposed system outperforms VINS-Loop. When viewpoint difference is not considered, such as in the previous subsection ($0\,\text{m}\backslash0°$), the RMSE of our algorithm is not better than VINS-Loop. It indicates that both algorithms can reduce drifts adequately if the MAV flies back to very similar places. As $\min_t$ and $\min_\psi$ grow larger, the differences become more noticeable. Until $\min_t$ and $\min_\psi$ reach about 3 m and 45°, VINS-Loop cannot provide reliable candidates anymore. However, the proposed algorithm can even close loops when the relative translation is over 7 m or the yaw angle exceeds 50°.

**Table 2.** Comparison of Loop Number and RMSE (m). The best result is highlighted in **bold**.

| $\min_t \backslash \min_\psi$ | VINS-Loop | | Proposed | |
|:---:|:---:|:---:|:---:|:---:|
| | **Loop Number** | **RMSE** | **Loop Number** | **RMSE** |
| $0\,\text{m}\backslash0°$ | 94 | 0.10 | 136 | 0.10 |
| $2\,\text{m}\backslash20°$ | 15 | 0.12 | 74 | **0.10** |
| $2.5\,\text{m}\backslash30°$ | 6 | 0.28 | 51 | **0.12** |
| $2.5\,\text{m}\backslash40°$ | 1 | 0.29 | 38 | **0.14** |
| $3\,\text{m}\backslash45°$ | 0 | 0.31 | 33 | **0.12** |
| $5\,\text{m}\backslash45°$ | 0 | 0.31 | 20 | **0.21** |
| $7\,\text{m}\backslash50°$ | 0 | 0.31 | 5 | **0.19** |

Some loop pairs with large viewpoint changes are shown in Figure 9, where $\theta_{ij}$ and $\phi_{ij}$ denote the relative pitch and roll angles. For example, the relative angle between the loop candidate and the current keyframe is over 50° in Figure 9a, and the relative translation is 7.4 m in Figure 9b. In such situations, our system can not only retrieve the correct loop candidate but also match features precisely. It is also worth noting that in Figure 9d, where the illumination condition is limited, the proposed method can also extract distinct descriptors to close the loop. Overall, it is convincing that our loop closure detection approach can perform more robustly against viewpoint changes due to its reliability of recalling and feature matching.

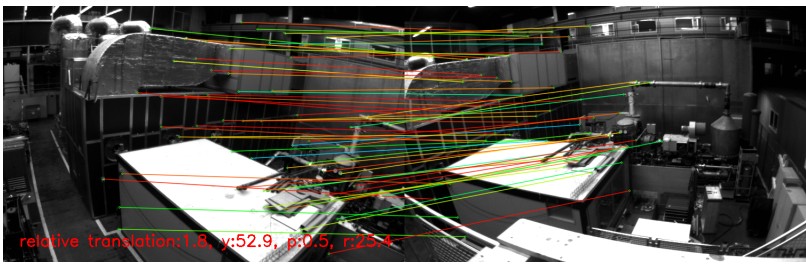

(**a**)

(**b**)

**Figure 9.** *Cont.*

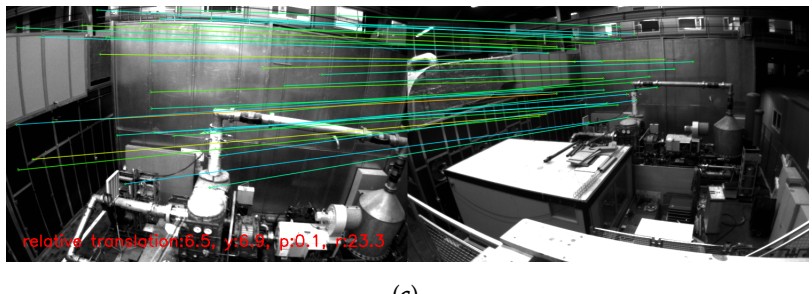

(**c**)

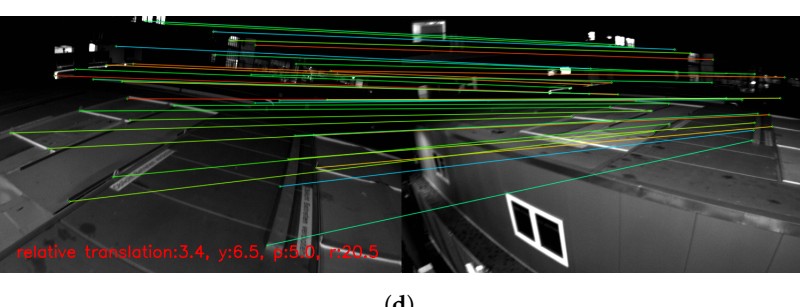

(**d**)

**Figure 9.** Examples of detected loop pairs with matched features at different viewpoints in EuRoC datasets. (**a**) $t_{ij} = 1.8$ m, $\psi_{ij} = 52.9°$, $\theta_{ij} = 0.5°$, $\phi_{ij} = 25.4°$. (**b**) $t_{ij} = 7.4$ m, $\psi_{ij} = 13.2°$, $\theta_{ij} = 8.6°$, $\phi_{ij} = 26.8°$. (**c**) $t_{ij} = 6.5$ m, $\psi_{ij} = 6.9°$, $\theta_{ij} = 0.1°$, $\phi_{ij} = 23.3°$. (**d**) $t_{ij} = 3.4$ m, $\psi_{ij} = 6.5°$, $\theta_{ij} = 5.0°$, $\phi_{ij} = 20.5°$

5.2.3. Multi-Session Performance

The EuRoC datasets contain several sessions for each of its three environments: five in Machine hall, three in Vicon room 1 and three in Vicon room 2. As the loop closure detection approaches also play an important role in merging multiple maps, the multi-session performance of the proposed system is validated. We sequentially process all the sessions corresponding to each environment. Each trajectory in the same environment has a ground truth with the same world reference, allowing a single global alignment to compute ATE. The first sequence in each room provides an initial map. Processing the following sequences starts with a new trajectory. It can be merged with the previous sequences when relocalized and optimized based on loop closure detection approaches.

Table 3 reports the global multi-session RMSE in the three rooms, compared with VINS-Loop. As for V201-V203 in Vicon room2, the proposed system obtains 1.3 better accuracy than VINS-Loop, and the advantage goes up to 2 times in multi-session. This improvement is mainly due to the frequent and robust loop detection in current and previous sequences. In Vicon Room 1, the RMSE of the whole trajectory is 0.055 m, about 6 cm better than VINS-Loop. Nevertheless, the difference in the results between our system and VINS-Loop is insignificant in MH01-05 sequences. We hypothesize that a greater depth scene for the machine hall sequences leads to richer features for loop closing, which is similar to in the single-session experiments.

**Table 3.** Multi-session RMSE (m) on the EuRoC dataset. The best result is highlighted in **bold**.

| Room | Sequences | VINS-Loop | Proposed |
|---|---|---|---|
| Vicon Room 1 | V101-03 | 0.061 | **0.055** |
| Vicon Room 2 | V201-03 | 0.162 | **0.086** |
| Machine Hall | MH01-05 | **0.075** | 0.087 |

*5.3. Real-World Experiments with a Hybrid Ground/Aerial Vehicle*

We conducted real-world experiments on a self-developed hybrid ground/aerial vehicle (Figure 10) to prove the effectiveness and robustness of our system on this special vehicle. The quad ducted-fan hybrid ground/aerial vehicle moves on the ground with

its continuous tracks and flies with four ducted fans. To sense the environment, it is also equipped with an Intel Realsense D435i camera and an NVIDIA Jetson Xavier NX. The laboratory for the indoor experiments is shown in Figure 11. The robot's ground-truth position is provided by a NaturalPoint OptiTrack motion capture system, whose effective range in this room is about 5 m. During the experiments, the hybrid ground/aerial vehicle was manually controlled. It reached about 1.2 m after take-off and flew three rounds stably. After landing on the ground, it drove nearly one round along a circular path with continuous tracks. Color images from the left camera (15 Hz) and IMU measurements (200 Hz) of D435i were recorded on-board and stored in rosbags.

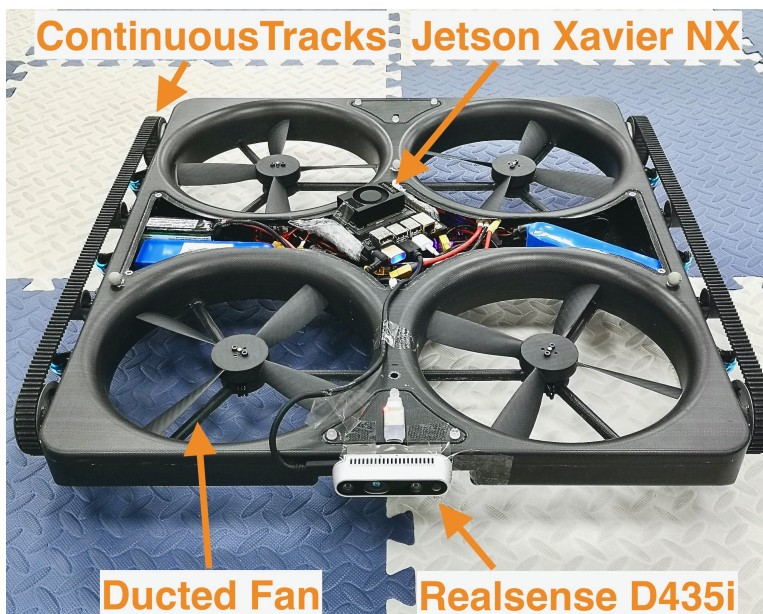

**Figure 10.** A self-developed hybrid ground/aerial vehicle with continuous tracks, ducted fans, an Intel Realsense D435i camera and an NVIDIA Jetson Xavier NX.

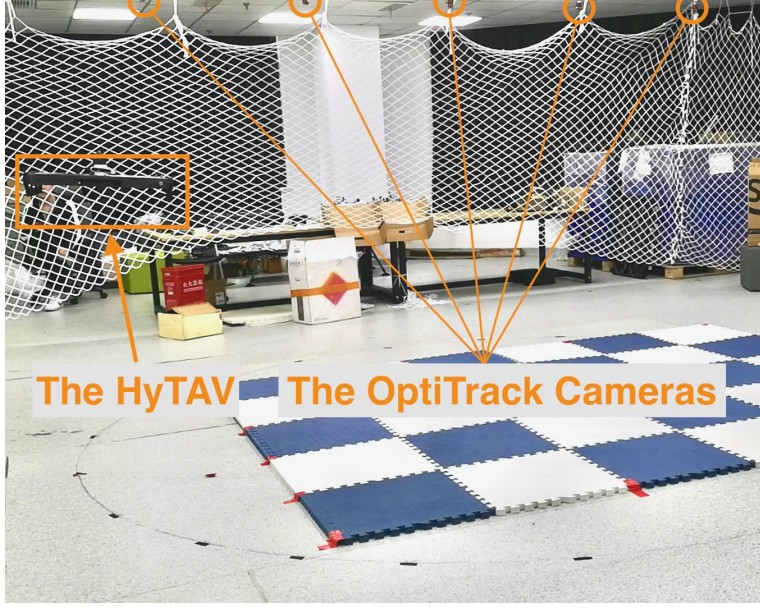

**Figure 11.** A laboratory with the NaturalPoint OptiTrack motion capture system.

We compare our algorithm with VINS-Loop, as both ours and VINS-Loop use the tracking results of VINS-VIO. The results are shown in Table 4. Because of the high speed the vehicle flew at and the inaccurate IMU data, VINS-VIO suffers noticeable drifts in

the experiments. Specifically, the RMSE of VINS-VIO is 0.18 m. Meanwhile, VINS-Loop reduces the error of VINS-VIO slightly (only 0.015 m better), since only a few loops have been detected. In contrast, the proposed system achieves about 1.5 times more accurate results than VINS-VIO, and the RMSE is 0.04 m lower than VINS-Loop.

**Table 4.** RMSE (m) in the Indoor Experiment. The best result is highlighted in **bold**.

| System | RMSE (m) | Reduced Error |
|---|---|---|
| VINS-VIO | 0.18 | - |
| VINS-Loop | 0.165 | 8.3% |
| Proposed | **0.125** | **30.5%** |

To illustrate the detected loops in the experiments, in Figure 12, we display the trajectories with loop links of VINS-Loop and ours, respectively. Each trajectory has two parts: three flying rounds (the upper zigzag circles) and a driving round (the lower smooth circle). Intuitively, the proposed system has drastically more loop links than VINS-Loop. In Figure 12a, only several loops with similar views are closed, and almost no loops between aerial and ground-level views are detected. On the contrary, plentiful loops with large viewpoint changes can be observed in Figure 12b.

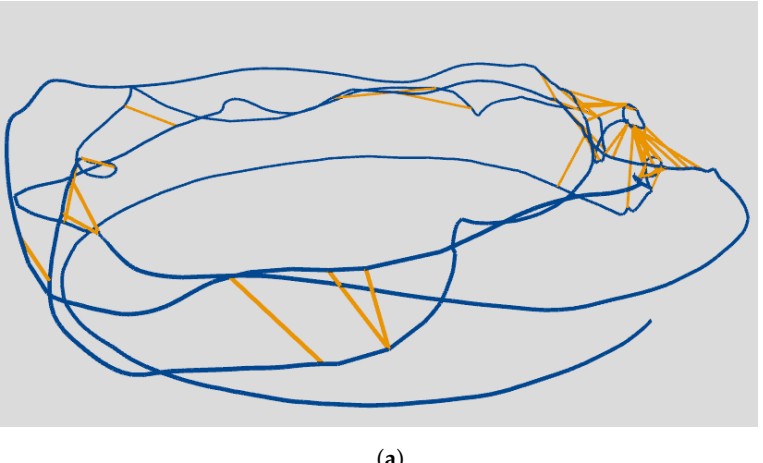

(**a**)

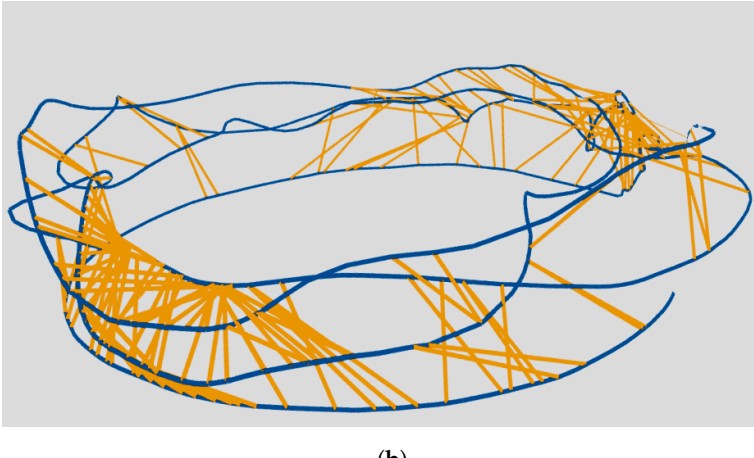

(**b**)

**Figure 12.** Trajectories of the real-world experiment with loop links (yellow lines). (**a**) VINS-Loop. (**b**) Proposed.

Some examples of the detected loop pairs with large view variance are shown in Figure 13. The symbols of relative translation and angles are the same as in the experiments on datasets. We can observe apparent image differences despite small relative translation

and angles since the depth scene in the laboratory is obviously short. We can see that the two left images were captured at the ground-level view, while the two loop candidates on the right were taken in the air. A change in viewpoint not only altered the scale of the pictures, but also the illumination of the features. Even so, the loop is correctly detected, and features are matched precisely. According to the experimental results in the real world, it is evident that the proposed system fairly solves the viewpoint change problem for hybrid ground/aerial vehicles.

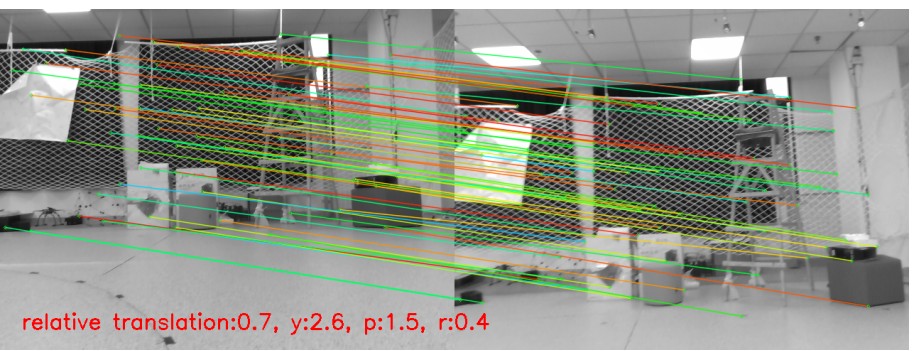

(**a**)

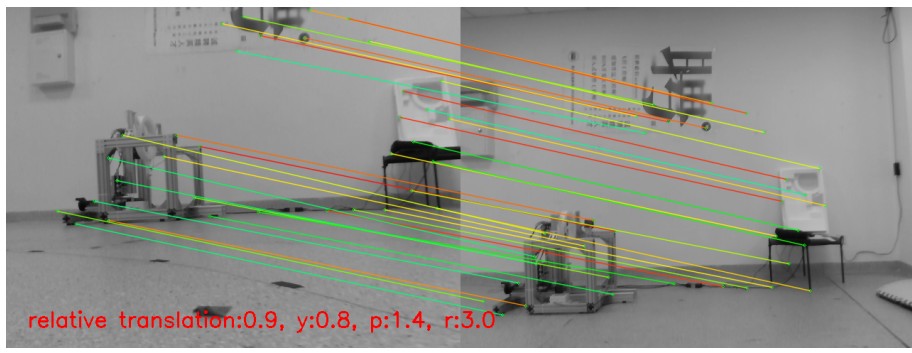

(**b**)

**Figure 13.** Examples of detected loop pairs at different viewpoints in the indoor experiment. (**a**) $t_{ij} = 0.7\,\mathrm{m}$, $\psi_{ij} = 2.6°$, $\theta_{ij} = 1.5°$, $\phi_{ij} = 0.4°$. (**b**) $t_{ij} = 0.9\,\mathrm{m}$, $\psi_{ij} = 0.8°$, $\theta_{ij} = 1.4°$, $\phi_{ij} = 3.0°$.

## 6. Discussion

In this paper, the problem of loop closure detection failures for HyTAVs caused by significant viewpoint changes is investigated. A method including offline vocabulary training and online loop closure detection is proposed. For offline vocabulary, the binary encoding approach is presented to speed up the image retrieval process of utilizing deep-learned descriptors in the traditional appearance-based approaches. For online loop closure detection, the SuperPoint features are extracted according to the keyframe's image and tracking keypoints, and the loop candidate is retrieved using the offline trained vocabulary. Finally, the constraint of the loop pair for graph optimization is computed by a PnP method based on the feature-matching results of SuperGlue.

We evaluate both the performance of the offline trained vocabulary as well as the effectiveness of the online approach on a public dataset and the dataset collected by a self-developed hybrid ground/aerial vehicle. First, the proposed approach of combing the BoW framework with an encoding method for vocabulary training and image retrieval is proven to be more effective than the handcrafted ORB and BRIEF descriptors, thanks to the distinctiveness of the binary-encoded descriptors. Moreover, according to the experimental results of online experiments, the system is shown to be able to outperform the state-of-the-art work VINS-Loop not only on the majority of EuRoC datasets, but also on our ground/aerial data. The performance margin of the proposed method over the VINS-Loop is larger in situations where there are large viewpoint changes, demonstrating that this

system is more robust to viewpoint changes. It is also proven that the proposed method has enabled detecting loops more frequently and reliably in the general case. Moreover, the algorithm runs in real-time with a state-of-the-art SLAM framework on a modern CPU and GPU.

Although the approach is proposed for hybrid ground/aerial vehicles, it can also be expanded to UAVs, as they also suffer from viewpoint variances. Moreover, according to the experiments in multi-session, we find its potential in collaborative multi-robot systems, which need the coordination of different vehicles by loop closure. However, the swarms of UAVs or hybrid ground/aerial vehicles have high demand on communication bandwidth. To reduce the demand on the bandwidth, it would be of interest to explore the approach with feature descriptors that are not only distinctive but also compact.

**Author Contributions:** Conceptualization, Y.W. and B.X.; methodology, Y.W. and W.F.; software, Y.W.; validation, Y.W., W.F. and B.X.; investigation, Y.W.; resources, W.F.; writing—original draft preparation, Y.W.; writing—review and editing, B.X.; visualization, W.F.; supervision, C.X.; project administration, C.X.; funding acquisition, C.X. All authors have read and agreed to the published version of the manuscript.

**Funding:** This research was funded by the National Natural Science Foundation of China (No. 52102432), the National Key Research and Development Project of China (No. 2020YFC1512500) and the National Natural Science Foundation of Chongqing (No. cstc2020jcyj-msxm3857).

**Data Availability Statement:** Data sharing not applicable.

**Conflicts of Interest:** The authors declare no conflict of interest.

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
