# Peer review of "A Robust and Efficient Loop Closure Detection Approach for Hybrid Ground/Aerial Vehicles"

_drones, doi:10.3390/drones7020135_

Round 1

Reviewer 1 Report

This paper presents research on how to handle loop closure, one of the important topics of SLAM, more robustly and efficiently.

Considering both ground/aerial vehicles, performance verification was performed based on representative datasets such as EuRoC and actual data acquired through specimens, and it shows better performance compared to other studies.

In general, it is well organized, but there are some parts that need some supplementation.

1. (Line 89) It is necessary to add the reference of UCN and DeepDesc.

2. (Chapter 3) In Sections 3.1 and 3.2 of Chapter 3, it would be good to add an overview paragraph for each section.

3. (Eq. 1) typo error

Author Response

We would like to thank the reviewer for providing the constructive review comments.

Point1: (Line 89) It is necessary to add the reference of UCN and DeepDesc.

Response1: We apologize for missing the references in the original manuscript. Now we have added them. (line 91)

Point2: (Chapter 3) In Sections 3.1 and 3.2 of Chapter 3, it would be good to add an overview paragraph for each section.

Response2: Thank the reviewer for the advice. We have now added an overview paragraph for Section 3.1 (line 131-133) and Section 3.2 (line 194-196).

Point3:  (Eq. 1) typo error

Response3: We apologize for the typo and have now added the blank after if.

Reviewer 2 Report

Summary: This paper introduces a keypoint detection-based loop closure detection in applying hybrid ground/aerial vehicles. The key points extracted from SuperPoint network are used as tracking features to detect loop candidates with a BoVW-based method. The proposed framework is evaluated on EuRoC dataset and a real-world hybrid ground/aerial vehicles dataset. 

Pros:

1. The paper is easy to follow and well-motivated. I can easily understand the proposed framework and its block of methods. Using binary encoding to select the descriptors for the DBoW process is novel.

2. The ablation experiments are well-conducted. 

Cons: 

1. More details should be included in Figure 3. What is the meaning of different colors? 

2. More quantitative results should be conducted with different deep learning-based encoders for descriptors. The current comparison mainly focuses on those non-deep-learning-based methods for keypoint detections. I am wondering if there is a better option for keypoint detection in this application.

Overall this paper is good. I tend to accept this paper with minor revisions.

Reviewer 3 Report

Some comments and suggestions for the authors of the paper are included in the attachment.
